# Human aminolevulinate synthase structure reveals a eukaryotic-specific autoinhibitory loop regulating substrate binding and product release

Henry J. Bailey[1,5], Gustavo A. Bezerra[1,5], Jason R. Marcero[2,5], Siladitya Padhi [3,5], William R. Foster [1], Elzbieta Rembeza[1], Arijit Roy [3], David F. Bishop[4], Robert J. Desnick[4], Gopalakrishnan Bulusu [3], Harry A. Dailey Jr[2] & Wyatt W. Yue [1✉]

5′-aminolevulinate synthase (ALAS) catalyzes the first step in heme biosynthesis, generating 5′-aminolevulinate from glycine and succinyl-CoA. Inherited frameshift indel mutations of human erythroid-specific isozyme ALAS2, within a C-terminal (Ct) extension of its catalytic core that is only present in higher eukaryotes, lead to gain-of-function X-linked protoporphyria (XLP). Here, we report the human ALAS2 crystal structure, revealing that its Ct-extension folds onto the catalytic core, sits atop the active site, and precludes binding of substrate succinyl-CoA. The Ct-extension is therefore an autoinhibitory element that must re-orient during catalysis, as supported by molecular dynamics simulations. Our data explain how Ct deletions in XLP alleviate autoinhibition and increase enzyme activity. Crystallography-based fragment screening reveals a binding hotspot around the Ct-extension, where fragments interfere with the Ct conformational dynamics and inhibit ALAS2 activity. These fragments represent a starting point to develop ALAS2 inhibitors as substrate reduction therapy for porphyria disorders that accumulate toxic heme intermediates.

[1] Structural Genomics Consortium, Nuffield Department of Medicine, University of Oxford, Oxford OX3 7DQ, UK. [2] Department of Biochemistry and Molecular Biology, University of Georgia, Athens, GA 30602, USA. [3] TCS Innovation Labs-Hyderabad (Life Sciences Division), Tata Consultancy Services Ltd, Hyderabad 500081, India. [4] Department of Genetics and Genomics Sciences, Icahn School of Medicine at Mount Sinai, New York, NY 10029, USA. [5]These authors contributed equally: Henry J. Bailey, Gustavo A. Bezerra, Jason R. Marcero, Siladitya Padhi. ✉email: wyatt.yue@sgc.ox.ac.uk

The tetrapyrrole cofactor heme is essential for various cellular processes across lifeforms. In metazoans, heme synthesis occurs via a conserved eight-reaction pathway that requires iron, glycine, and succinyl-CoA. The first and rate-limiting step is carried out by 5′-aminolevulinate synthase (ALAS; EC 2.3.1.37) in the mitochondria[1,2]. ALAS catalyses the pyridoxal 5′-phosphate (PLP)-dependent condensation of succinyl-CoA and glycine to form aminolevulinic acid (ALA), with CoA and $CO_2$ as by-products. The product ALA is then transported into the cytoplasm for the subsequent biosynthetic steps that eventually lead back into the mitochondria, where ferrochelatase (FECH) ultimately inserts iron into protoporphyrin IX (PPIX) to produce heme[3,4]. Vertebrates have evolved two nuclear-encoded ALAS isozymes[5,6] that share ~60% amino acid sequence identity. ALAS1 (gene location 3p21.2) is the housekeeping enzyme providing a basal level of heme in non-erythroid cell types for cytochromes and other hemoproteins. ALAS2 (gene location Xp11.21) is predominantly expressed in erythroid progenitor cells, and synthesizes 85–90% of total body heme specifically for hemoglobin production during erythropoiesis. Mutations in the heme biosynthetic genes, resulting in an accumulation of toxic porphyrin intermediates, are associated with a group of inherited disorders called porphyrias[3,7].

The ALAS catalytic mechanism, involving Schiff-base formation between the aldehyde carbonyl of the PLP cofactor and ε-amino group of a conserved lysine residue, has been well characterized[8–11]. Without substrates, PLP is covalently attached to the active site lysine (Lys391 in human ALAS2 (hsALAS2)) as an internal aldimine adduct. The reaction begins with the glycine substrate replacing lysine in the Schiff-base linkage to form an external aldimine. Next, deprotonation of glycine forms a transient quinonoid intermediate that is competent for nucleophilic attack on the co-substrate, succinyl-CoA, resulting in a condensation reaction releasing the CoA moiety, followed by a decarboxylation assisted by the active site histidine (His285 in hsALAS2). Subsequent protonation regenerates the internal aldimine, and allows release of the product, ALA. A number of kinetic studies have suggested that the ALAS rate-determining step is product (ALA) release[8,9]. The structure of R. capsulatus ALAS (rcALAS) was the first reported for the enzyme[12], revealing an induced-fit mechanism upon substrate binding, via an open-to-close transition of a mobile active site loop. Conformational mobility of this loop has also been proposed to be a kinetic barrier for product release[8,11,13].

Eukaryotic ALAS enzymes have evolved extensions appending to both N- and C-termini of the highly conserved catalytic core[14] (Fig. 1a). The N-terminal extensions, harboring the mitochondrial targeting sequence[14–17], are poorly conserved between higher and lower eukaryotic ALAS enzymes, as well as between metazoan ALAS1 and ALAS2 (Supplementary Fig. 1). Metazoan enzymes further encode three Cys-Pro motifs[18], two of which are within the targeting sequence. The third Cys-Pro motif, together with an invariant Gln-Glu-Asp-Val motif, are found within a 70–90 aa region of poor sequence conservation and unknown function. In both ALAS1 and ALAS2, the Cys-Pro motifs have been shown to be responsible for heme-dependent inhibition of mitochondrial translocation of the enzyme precursor[19,20].

The eukaryotic extension at the C-terminus (Ct-extension) ranges from 35 to 60 aa in length. Metazoan ALAS1 and ALAS2 share ~53% sequence identity in their Ct-extensions (their major differences are found in the last 9 amino acids), while S. cerevisiae encodes an entirely different Ct-extension from metazoans, and also from S. pombe (Supplementary Fig. 1). Frameshift indel mutations in exon 11 of the human ALAS2 gene (on chromosome Xp11.21) that result in deletion, replacement, or elongation of its Ct-extension are the molecular cause of X-linked protoporphyria (XLP,

MIM 300752), an inherited disorder that presents with painful phototoxicity and an increased risk for liver dysfunction and failure[21,22], due to high levels of the toxic heme intermediate PPIX. At the protein level, these genetic lesions cause enhanced enzyme activity, lending to XLP being referred to as a gain-of-function (GOF) disorder[23–25]. Recently, a GOF XLP phenotype was reported for mutations of the mitochondrial ATP-dependent Clp protease ClpX, an AAA+ (ATPases associated with diverse cellular activities) unfoldase shown to activate ALAS dimers through partial unfolding of the enzyme to enhance loading of the PLP cofactor into the active site[26,27]. XLP contrasts with another ALAS2-associated blood disorder, X-linked sideroblastic anemia (XLSA, MIM 300751), which results in loss of enzyme function and is characterized by heme-deficient and iron-overloaded red blood cells (ringed sideroblasts)[28–31]. XLSA is attributable to mutations within exons 5–11 (including a few in the Ct-extension; from Human Gene Mutation Database)[32] that are predominantly missense in nature.

While the dichotomous genotype-phenotype correlation of XLSA and XLP is now widely recognized, it remains unknown how the GOF phenotypes in XLP are brought about by various mutations in the ALAS2 Ct-extension, although a self-inhibitory role has been proposed for this region[33]. The recently published ALAS2 structure from S. cerevisiae (scALAS) showed each Ct-extension wrapping around the surface of the ALAS homodimer, albeit distant from its active site[34]. This sequence, furthermore, bears no homology with other eukaryotic counterparts and is likely to be evolutionarily distinct from the mammalian ALAS2 Ct-extension. Importantly, deletion of the scALAS extension does not increase its enzyme activity[34], contrary to the XLP-causing mutations in hsALAS2. Here, we determine the crystal structure of hsALAS2 which, in combination with molecular dynamics simulations and enzyme kinetics, reveals salient features of the Ct-extension that explain its role in enzyme regulation and in causing XLP.

## Results

**hsALAS2 without presequence is sufficient for catalysis**. We expressed in E. coli, as soluble proteins, several constructs of hsALAS2 with different truncations within the N-terminal extension, while keeping their Ct-extensions intact (Supplementary Fig. 2). The isolated as-purified proteins showed varying degrees of proteolysis (see below), as observed previously[24], which we attributed to cleavage of the 40-aa Ct-extension. For crystallography experiments, we next employed baculovirus-infected insect Sf9 cells to express hsALAS2 aa 79–587 (hsALAS2$_{\Delta N78}$) and aa 143–587 (hsALAS2$_{\Delta N142}$) resulting in soluble proteins with no signs of proteolytic degradation. The majority of the as-purified hsALAS2$_{\Delta N78}$ and hsALAS2$_{\Delta N142}$ proteins from insect cells contain the PLP cofactor, with their Schiff base predominantly in the deprotonated enolimine tautomer[35] (Supplementary Fig. 3). Insect cell-expressed hsALAS2 is active with $K_m$ values (31 ± 4 μM for succinyl-CoA, 11.8 ± 1.0 μM for glycine) close to those reported by others using a similar assay[24] (Supplementary Fig. 4a, b). This form of the enzyme is also capable of forming an interaction with the previously reported partner, beta subunit (SUCLA2) of the succinyl-CoA synthetase complex SUCLG1-SUCLA2 (refs. [36,37]), as revealed by cross-linking using disuccinimidyl glutarate (Supplementary Fig. 4c).

**Overall structure of hsALAS2 and conserved features**. We first crystallized the hsALAS2$_{\Delta N78}$ protein and determined its crystal structure at 2.7 Å resolution by molecular replacement, using the rcALAS structure as a search model. Unexpectedly, the N-terminal residues 79–142 of our construct were not visible in the electron density. Intact mass spectrometry of purified hsALAS2$_{\Delta N78}$

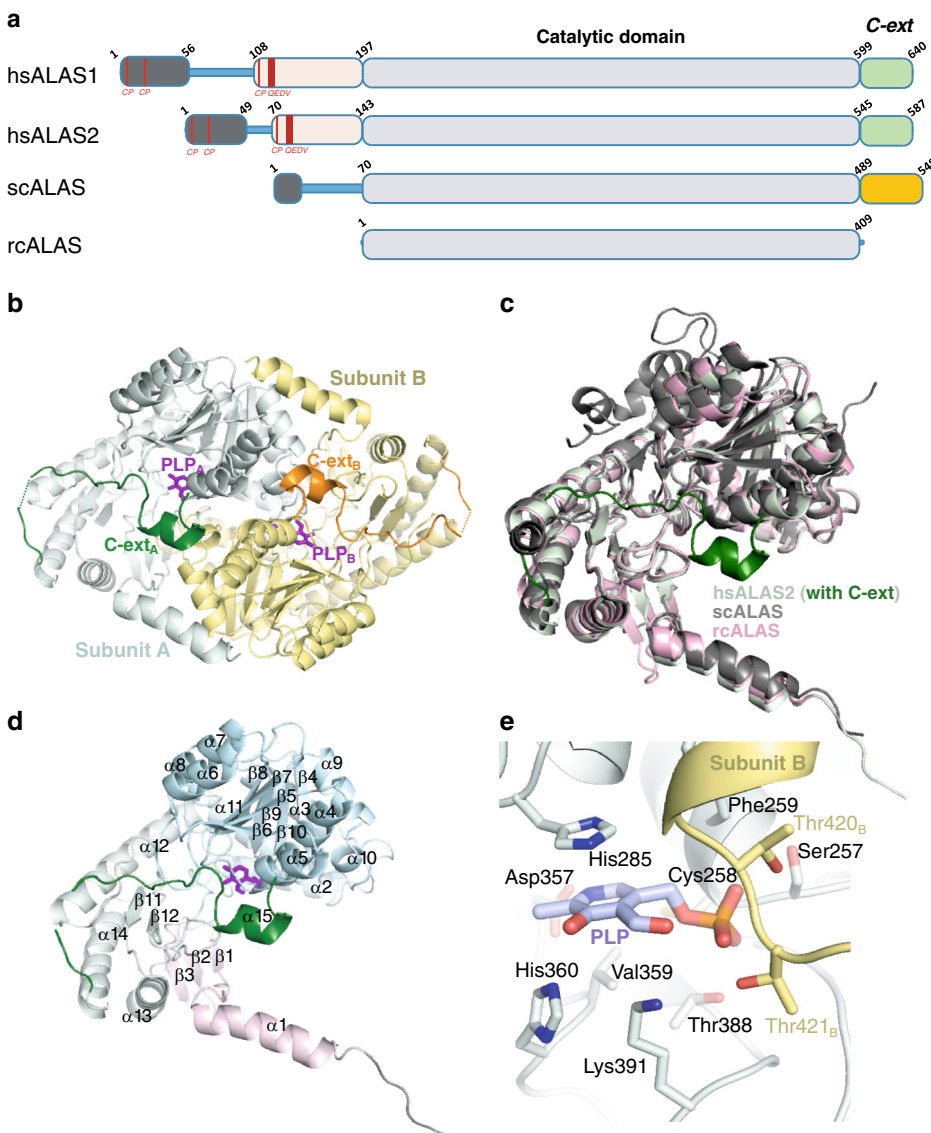

**Fig. 1 Domain organization and structure of hsALAS2. a** Domain architecture of hsALAS1, hsALAS2, scALAS, and rcALAS, highlighting the catalytic core (gray box) flanked by N-terminal (black, pink boxes) and C-terminal extension (green in higher eukaryotes, orange in lower eukaryotes). **b** hsALAS2 homodimer (this study) composed of monomer A (catalytic domain in gray, Ct-extension in dark green) and the opposite monomer B (catalytic domain in yellow, Ct-extension in orange). PLP is shown in purple sticks. **c** Structure superimposition of protomer from hsALAS2 (this study), scALAS (PDB 5TXR) and rcALAS (PDB 2BWN). **d** Domain organization and secondary structure assignment for hsALAS2. Subdomain 1 is shown in pink, subdomain 2 in gray, subdomain 3 in cyan, and the Ct-extension in dark green. **e** PLP binding site of hsALAS2. PLP-interacting residues from monomer A are shown in gray and from opposing monomer B in yellow. PLP is shown in mauve (carbon color).

nonetheless indicates that aa 79–142 are present (Supplementary Fig. 5a), suggesting this region remains intact in the crystal, albeit disordered. This is further revealed by crystal packing analysis of ALAS2 protomers in the crystal lattice, leaving space for the N-terminal region (Supplementary Fig. 5b). Subsequently, the shorter hsALAS2$_{\Delta N142}$ protein was crystallized, and its higher resolution structure at 2.3 Å resolution was determined (Table 1) and will be referred to hereafter in the manuscript.

The crystal asymmetric unit contains an hsALAS2 homodimer (Fig. 1b), as shown in solution by size exclusion chromatography and small angle X-ray scattering (SAXS)(Supplementary Fig. 6, Supplementary Table 1). The two monomers (rmsd 0.167 Å) are tightly interlocked in the structure, burying a total of 4850 Å$^2$ accessible surface per monomer. The dimer interface is largely derived from the catalytic domain and highly conserved with the rcALAS structure (Supplementary Fig. 1), although the

Ct-extension contributes three aromatic residues that mediate inter-monomer interactions (see next section). This said, an hsALAS2 construct encoding the catalytic domain alone (aa 143–545, hsALAS2$_{\Delta N142\Delta C545}$) is sufficient to homodimerize in solution (Supplementary Fig. 6b, e).

The catalytic core of hsALAS2 is highly similar to rcALAS (sequence identity 50%, rmsd 1.1 Å) and to scALAS (45%, 1.5 Å) (Fig. 1c). Together they are part of the superfamily of type I PLP-dependent enzymes, in which the catalytic core can be dissected into three sub-domains. In hsALAS2 (Fig. 1d), they correspond to: subdomain 1 (aa 143–195) forming an α-helix and a three-stranded, antiparallel β-sheet; the large central subdomain 2 (aa 196–439) forming a seven-stranded β-sheet enclosed by nine α-helices in repeated β/α motifs; and the last subdomain 3 (aa 440–544) including a three-stranded antiparallel β-sheet, three α-helices and a fourth β-strand that contributes to the sub-domain 1 β-sheet.

**Table 1 Data collection and refinement statistics for hsALAS2$_{\Delta N142}$ and hsALAS2$_{\Delta N78}$ crystal structures.**

| Dataset | hsALAS2$_{\Delta N142}$ | hsALAS2$_{\Delta N78}$ |
|---|---|---|
| Beamline | Diamond beamline I04 | Diamond beamline I24 |
| Wavelength (Å) | 0.9795 | 0.9795 |
| Unit cell parameters (a, b, c) | 125.77 Å, 107.7 Å, 75.71 Å | 78.21 Å, 155.62 Å, 86.24 Å |
| ($\alpha, \beta, \gamma$) | 90.0°, 109.0°, 90.0° | 90.0°, 98.45°, 90.0° |
| Space group | C 1 2 1 | P 1 21 1 |
| Resolution range (Å) | 71.53-2.30 | 85.32-2.65[a] |
| Unique reflections | 40,141 | 58,818 |
| Rsym (%) | 0.09 (0.71) | 0.21 (0.76) |
| I/sig(I) | 5.8 (1.0) | 4.9 (1.5) |
| Completeness (%) | 94.8 (95.0) | 72.6 (19.3)[a] |
| Multiplicity | 2.3 (2.3) | 3.4 (3.0) |
| $R_{cryst}$ (%) | 0.21 | 0.21 |
| $R_{free}$ (%) | 0.23 | 0.26 |
| Wilson B factor (Å$^2$) | 43.16 | 26.8 |
| Average total B factor (Å$^2$) | 55.2 | 25.8 |
| R.m.s.d. bond length (Å$^2$) | 0.003 | 0.004 |
| R.m.s.d. bond angle (°) | 0.69 | 0.68 |
| Clashscore | 6.7 | 5.78 |
| Ramachadran favored (%) | 95.6 | 95.6 |
| Ramachandran disallowed (%) | 0 | 0 |
| Rotamer outliers (%) | 0 | 0.15 |

Data from highest resolution shell shown in parenthesis.
[a]Anisotropic data truncated in STARANISO using local I/sigI cut off at 1.2 results in the inclusion of data to 2.65 with low isotropic completeness.

Like its orthologs, hsALAS2 is an obligate homodimer in that each active site is formed from both monomeric subunits, with one PLP molecule bound per active site. Prior to crystallization, the protein was treated with hydroxylamine to convert the covalently-bound PLP (in equilibrium between two tautomers; Supplementary Fig. 3) into the non-covalent form that is more homogenous (Supplementary Fig. 7). As expected from the sequence alignment (Supplementary Fig. 1), hsALAS2 employs a highly conserved set of interactions with PLP, involving mainly the central sub-domain of the catalytic core (Fig. 1e). In short, the pyridinium ring fits snugly through hydrophobic interactions with His285 and Val359, as well as hydrogen bonds with His360 and Asp357. The PLP phosphate is neutralized by four hydrogen bonds from its own subunit (Ser257 and Thr388 side-chains, Cys258 and Phe259 main-chains), as well as three hydrogen bonds contributed from the opposing monomer (Thr420 and Thr421 side-chains, Thr421 main-chain).

**hsALAS2 C-terminus is a self-inhibitory loop.** The largest structural difference between hsALAS2 and homologs resides in the C-terminus following the catalytic core. The 40-residue hsALAS2 Ct-extension (Fig. 1b and Supplementary Fig. 8) begins with a poorly structured segment (aa 544–557) which stretches outward to the protein exterior. There is no clear electron density for part of this segment (aa 549–555, Supplementary Fig. 8a), suggesting its mobility. The next segment of the Ct-extension (aa 560–568, Supplementary Fig. 8b) threads across the interface of sub-domains 2 and 3 as a linear polypeptide, with several main-chain amides and carbonyls shielded by hydrogen-bonding with catalytic core residues (Arg479 and Glu513, which are not present in scALAS or rcALAS). A two-turn helix (α15; Ser568-Phe575)

then ensues, forming a lid atop the PLP-bound active site, before being directed to the protein exterior (Phe575-Met578). Close examination of helix α15 (Fig. 2a) reveals that its two Glu residues, Glu569 and Glu571, form charged interactions respectively with Arg511 from subdomain 3 and Lys299/Arg293 from subdomain 2. This helix also contributes three aromatic residues Trp570, Tyr574, and Phe575 which stack among themselves through π-π interactions, and anchor the dimerization interface through hydrophobic interactions with Phe267, Thr268, and Ala415 of the opposite dimer subunit. Additional hydrogen bonds were contributed from Phe575 and Gly576 (main chain), with Lys271 on the opposing subunit and Arg293, respectively. The last 9 residues of the hsALAS2 Ct-extension (aa 579–587) were not modeled in the structure, likely due to high disorder as they are surface-exposed. These residues also represent one of the largest sequence divergences between ALAS1 and ALAS2 (Supplementary Fig. 1).

The succinyl-CoA-bound rcALAS structure (PDB 2BWO)[12] revealed residues involved in substrate succinyl-CoA binding, which are nearly invariant across the phyla (based on sequence alignment in Supplementary Fig. 1), suggesting that hsALAS2 has evolved the same binding site. However, when superimposing hsALAS2 and succinyl-CoA-bound rcALAS structures, we observed to our surprise that the hsALAS2 succinyl-CoA site is blocked by helix α15 and the C-terminus (Fig. 2b), which would sterically overlap with the ADP-diphosphate moiety of a modeled succinyl-CoA molecule. Our hsALAS2 structure therefore represents a conformation of the Ct-extension that precludes binding of succinyl-CoA to the active site. The steric blockade is mediated by several interactions between helix α15 and the catalytic core (Fig. 2a). These include (i) salt bridges of Glu571 with both Lys299 and Arg293, blocking the expected position of the succinyl-CoA ADP adenine group, (ii) the hydrogen bond between Gly576 (main-chain) and Arg293 that blocks the 3′-phosphoadenosine moiety, and (iii) the salt bridge between Glu569 and Arg511. It is of note that Arg511, a residue from the mobile active site loop (aa 505–514), also forms a salt bridge with Asp159 from helix α1. In rcALAS, the equivalent Arg511:Asp159 interaction plays a role in succinyl-CoA-induced conformational change during catalysis[12] (cf. Figure 3a).

The observed direct interaction between the Ct-extension and the active site loop in our hsALAS2 structure, through Arg511, implies that the mobility of the active site loop is driven by a conformational change of the Ct-extension, and suggests that these two regions act in concert to drive the ALAS2 catalytic cycle of substrate binding and product release. We explored further the role of R511 by site-directed mutagenesis on the crystallized construct, hsALAS2$_{\Delta N142}$, to generate the R511E variant which abolishes charged interactions with Glu569 (Ct-extension) and Asp159 (active site loop). Compared to wild-type (WT), the R511E variant is more prone to proteolysis at the Ct-extension as shown by SDS-PAGE (Fig. 3b) and intact mass spectrometry (Supplementary Fig. 5c, d), suggesting that the interaction of the Ct-extesion with the catalytic domain active site loop has been significantly disrupted. Consistent with this, the R511E variant is >2 °C more thermolabile than WT protein, with melting temperature similar to the hsALAS2$_{\Delta N142\Delta C545}$ (WT or R511E) protein with Ct-extension omitted from the construct (Fig. 3c). Furthermore, we showed by steady-state kinetics that the proteolysis-prone R511E variant has 2-fold higher enzyme activity than WT, as indicated by $V_{max}$ and $k_{cat}$ values (Fig. 3d, e and Table 2). Of additional note, the R511E variant demonstrates a lower specificity ($k_{cat}/K_m$) for glycine than WT protein, while the opposite was observed for succinyl-CoA (Table 2). Altogether, our data suggest that the salt bridge network of Glu569:Arg511:Asp159 plays an important role in regulating the Ct-extension during catalysis.

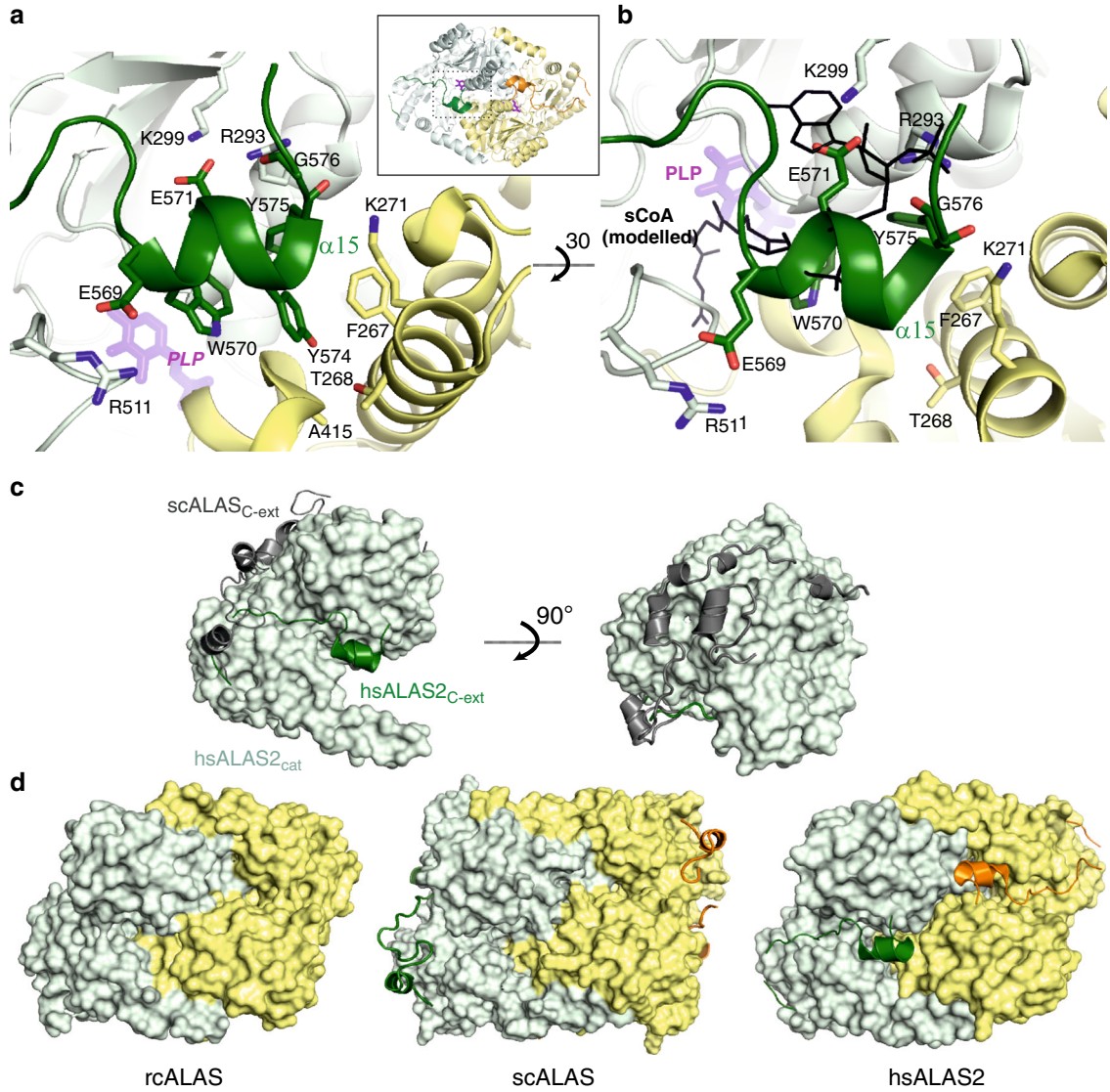

**Fig. 2 hsALAS2 C-terminal extension is mobile and interferes directly with active site. a** Interactions of helix α15 from the Ct-extension (dark green) with the catalytic core of monomer A shown in light green and the opposing monomer B shown in yellow. **b** Similar view as panel A, related by 30° rotation along the horizontal axis. The succinyl-CoA (sCoA) ligand, modeled into hsALAS2 from rcALAS (PDB 2BWO), is shown in black lines to highlight steric clashes with helix α15. **c** hsALAS2 monomer with catalytic domain in surface representation (light green) and Ct-extension in cartoon representation (dark green). The scALAS Ct-extension is also superimposed in cartoon representation (gray). **d** rcALAS (PDB 2BWN), scALAS (PDB 5TXR) and hsALAS2 (this study) homodimers. Catalytic domains are shown in surface representation (monomer A light green, monomer B yellow). Ct-extensions are shown in cartoon representation for scALAS and rcALAS (dark green for monomer A, orange for monomer B).

Comparison of our hsALAS2 structure with that of scALAS[34] shows that their Ct-extensions not only differ in sequence (Supplementary Fig. 1) but also in structural locations and conformations (Fig. 2c, d). The point of structural divergence occurs after the last helix (α14) in the catalytic domain, at position Leu545 of hsALAS2 (Leu489 of scALAS). The hsALAS2 extension threads across the subdomain interface and shields the substrate binding site from the exterior (Fig. 2c, left). In scALAS, by contrast, a two-turn helix directs its Ct-extension back along the periphery of sub-domain 3, packs against a unique 22-aa insertion in sub-domain 2 (aa 302–323, scALAS numbering) not present in hsALAS2 (Fig. 2c, right), and continues to wrap around the homodimer surface reaching as far as the N-terminal sub-domain 1 of the opposing monomer. Unlike hsALAS2, each Ct-extension in scALAS did not interact with its substrate binding site, leaving it exposed as in the rcALAS structure (Fig. 2d).

**Simulations of the ALAS2 C-terminal dynamics.** We next employed molecular dynamics (MD) simulations to understand the mobility of the hsALAS2 Ct-extension. Based on the experimental structure, we modeled hsALAS2 in the *apo* form, the *holo* form with PLP covalently-bound to Lys391, and the ternary complex with glycine-PLP and succinyl-CoA. In the substrate-free *apo* and *holo* forms, the Glu569:Arg511 and Glu571:Arg293/Lys299 salt bridges, serving to stabilize the Ct-extension over the active site, remain intact for most of the duration of the simulations (Fig. 4a, left and Supplementary Figs. 9–11). In order for succinyl-CoA to bind and orientate in the active site, we reason that the Ct-extension must change its conformation. Indeed, simulation of the substrate-bound ternary complex shows complete breakage of the salt bridges (Fig. 4a, right and Supplementary Figs. 9–11), resulting in an open conformation that is structurally poised to bind the substrate succinyl-CoA (Fig. 4b and Supplementary Movies 1 and 2). We also carried out MD

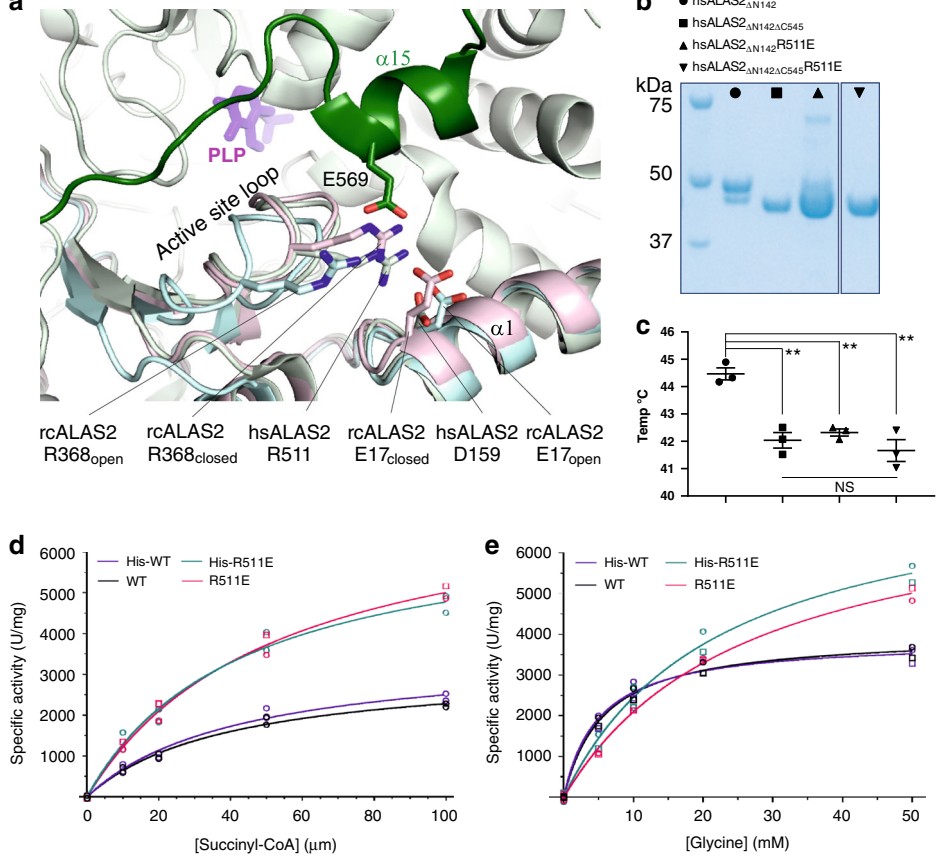

**Fig. 3 Role of C-terminal extension in structural and functional integrity. a** View of the hsALAS2 active site loop conformation within one monomer (catalytic domain light green, Ct-extension dark green). The active site loop from the open and closed forms of rcALAS (PDB 2BWN) are overlayed onto hsALAS2. **b** SDS-PAGE of the purified samples from hsALAS2$_{\Delta N142}$ WT (square) and R511E variant (up triangle), as well as hsALAS2$_{\Delta N142\Delta C545}$ WT (circle) and R511E variant (down triangle). **c** Midpoint melting temperatures, determined by differential scanning fluorimetry, of the four samples in panel **b**. Error bars represent mean values ±SEM from $n = 3$ technical replicates. **p-values determined by two sample $t$-test; $p = 0.0026$, 0.0011, and 0.0002 from left to right. **d, e** Progression curves for recombinant hsALAS2$_{\Delta N142}$ WT and R511E variant, for both His-tagged and untagged proteins. Specific activity is equivalent to the initial velocity normalized to protein in each assay. Succinyl-CoA and glycine titrations contained 50 mM glycine and 100 μM succinyl-CoA, respectively. Each nonlinear regression line was fitted to data from two biological replicates ($n = 2$), with the data for each replicate (average of two technical replicates) plotted as distinct symbols (circle and square). Source data for Fig. 3b–e are provided as a Source Data file.

| Table 2 Steady-state kinetics of recombinant hsALAS2 to study effect of His-tag and variants. | | | | | | | |
|---|---|---|---|---|---|---|---|
| Enzyme (His$_6$-tagged or tag-free) | $k_{cat}$ (min$^{-1}$) | $V_{max}$, SCoA (U mg$^{-1}$) | $K_m$, SCoA (μM) | $k_{cat}/K_m$, SCoA (min$^{-1}$μM$^{-1}$) | $V_{max}$, Gly (U mg$^{-1}$) | $K_m$, Gly (mM) | $k_{cat}/K_m$, Gly (min$^{-1}$mM$^{-1}$) |
| WT | 3.0 ± 0.2 | 3200 ± 200 | 39 ± 6 | 0.075 ± 0.012 | 4000 ± 130 | 5.7 ± 0.7 | 0.52 ± 0.07 |
| His-WT | 3.2 ± 0.3 | 3600 ± 300 | 42 ± 8 | 0.076 ± 0.015 | 3900 ± 180 | 5.0 ± 0.9 | 0.64 ± 0.12 |
| R511E | 6.3 ± 0.5 | 7700 ± 500 | 53 ± 7 | 0.120 ± 0.019 | 7700 ± 300 | 26 ± 2 | 0.24 ± 0.03 |
| His-R511E | 6.4 ± 0.7 | 6800 ± 500 | 43 ± 8 | 0.15 ± 0.03 | 8000 ± 500 | 22 ± 3 | 0.29 ± 0.05 |

Kinetic parameters for recombinant N-terminal His$_6$-tagged and tag-free hsALAS2$_{\Delta N142}$ WT and R511E proteins. Reported parameters are best-fit values ±SE for progression curves (Fig. 3d, e) derived from the Prism software (GraphPad 8.0). Source data are provided as a Source Data file.

simulations on a hsALAS2 R511E model in which the salt bridge with Glu569 would not be formed, to support our experimental observations that Arg511 is important for the integrity of Ct-extension (see previous section and Fig. 3). Indeed, the accessibility of the Ct-extension in the R511E model is higher than WT, in that the R511E Ct-extension has moved away from the catalytic core over the course of the simulation, and this movement leads to an increase in the solvent accessible surface area (Supplementary Fig. 12).

During MD simulations, dimeric hsALAS2 exhibits asymmetry between the two subunits, a feature previously reported for murine ALAS2 (ref. [38]). This asymmetry is particularly observed in the dynamics of salt bridges from the two subunits of the ternary complex (Supplementary Figs. 10 and 11, compare bottom left and right panels). Asymmetry is also observed at the secondary structure level. Analysis of dynamic cross correlation maps (DCCM), which quantify correlations in motions between different regions of a protein[39], shows that covalent attachment of PLP to the active site

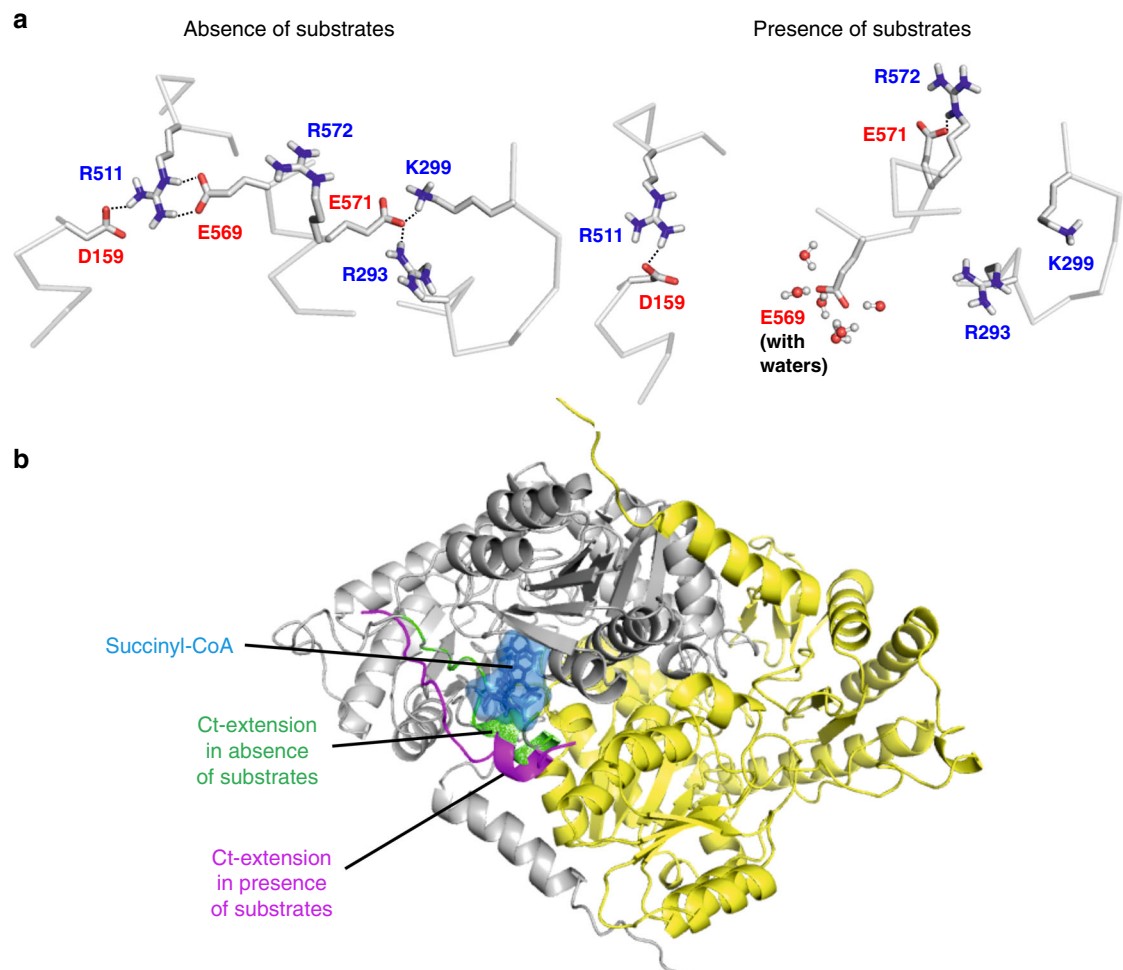

**Fig. 4 Dynamics of the Ct-extension revealed by MD simulations. a** Salt bridges in the absence (left) and presence (right) of substrates. The structure depicted on the left is the crystal structure subjected to energy minimization, and the structure depicted on the right is a representative snapshot taken from the simulation performed in the presence of substrates. Salt bridges are indicated by dashed lines. Water molecules stabililizing the E569 residue are shown in ball and stick representation. **b** The orientation of the Ct-extension in the absence and presence of substrates (shown in green and magenta, respectively). The space occupied by succinyl-CoA in the simulated structure is shown in blue.

leads to an increased anti-correlation between subdomain 2 (containing active site residues) of one subunit and subdomain 3 (containing the Ct-extension) of the other subunit (Supplementary Fig. 13a). The anti-correlation across the two subunits is facilitated by interaction of PLP from one subunit with Thr420 and Thr421 from the other subunit (Supplementary Fig. 13b). Altogether, the simulations data indicate the mobility of the hsALAS2 Ct-extension between different conformations during catalysis.

**Revised understanding of disease-causing mutations**. The hsALAS2 crystal structure provides a template to understand the molecular basis of disease-causing mutations for XLSA and XLP. To date, over 82 missense mutations affecting 69 different residues of hsALAS2 have been identified as causing XLSA (Supplementary Table 2 and Supplementary Fig. 14). These mutations are largely concentrated in the catalytic core and many have been previously mapped onto the rcALAS structure[12], allowing the categorization of their putative molecular defects including disruption of the dimerization interface, direct disruption of catalytic regions, or a global destabilization of the enzymatic fold.

The hsALAS2 Ct-extension sheds insights on some previously characterized mutations in the catalytic core, namely D159N and D159Y. The rcALAS open and closed structures (PDB 2BWN) show that formation and disruption of a salt bridge between Glu17

(Asp159 in hsALAS2) and Arg368 (Arg511 in hsALAS2) is key to transitioning between open and closed states (Fig. 3a). Aforementioned in our hsALAS2 structure, Arg511 forms a salt bridge to Glu569 of the Ct-extension, and to Asp159 in helix α1 of catalytic core (Fig. 3a). Results from our DSF, activity assays and MD simulations (Fig. 3b–e and Supplementary Fig. 12) altogether demonstrate the importance of the Arg511:Glu569 interaction towards the structural integrity of the Ct-extension. By contrast, disruption of the Asp159:Arg511 interaction, the likely culprit for the disease-causing D159N and D159Y variants, may reduce the ability of ALAS2 to adopt the catalytically active closed conformation, and stabilize the interaction of Arg511 with Glu569.

Gain-of-function XLP mutations all locate to exon 11 encoding the hsALAS2 Ct-extension. Three reported indel mutations (Q548X, delAT, and delAGTG) completely delete the two-turn α15 helical region that blocks the ALAS2 active site. A fourth known indel mutation (delG) leaves the Ct-extension helix intact, though it results in a frameshift and an additional 12-aa sequence downstream of that region[24] (Supplementary Fig. 2). Not surprisingly, the elongated C-terminus of recombinant delAGTG is subject to proteolysis during protein preparation, while the same region of the delG variant is not (Supplementary Fig. 15a). Disruption of the α15 helix would alleviate the restriction of conformational change imposed by the interaction of the

**Table 3 Steady-state kinetics of recombinant hsALAS2 XLP variants.**

| Enzyme (His$_6$-tagged) | $k_{cat}$ (min$^{-1}$) | $V_{max}$, SCoA (U mg$^{-1}$) | $K_m$, SCoA ($\mu$M) | $k_{cat}/K_m$, SCoA (min$^{-1}$ $\mu$M$^{-1}$) | $V_{max}$, Gly (U mg$^{-1}$) | $K_m$, Gly (mM) | $k_{cat}/K_m$, Gly (min$^{-1}$ mM$^{-1}$) |
|---|---|---|---|---|---|---|---|
| WT | 3.5 ± 0.2 | 3600 ± 200 | 32 ± 4 | 0.110 ± 0.014 | 3440 ± 90 | 8.5 ± 0.8 | 0.41 ± 0.04 |
| delAT | 15.4 ± 0.8 | 17,500 ± 800 | 56 ± 5 | 0.28 ± 0.03 | 16100 ± 310 | 10.3 ± 0.6 | 1.50 ± 0.12 |
| Q548X | 17.3 ± 1.1 | 19,900 ± 1000 | 55 ± 5 | 0.31 ± 0.04 | 17700 ± 330 | 12.4 ± 0.6 | 1.40 ± 0.11 |
| delG | 4.5 ± 0.2 | 4000 ± 180 | 20 ± 3 | 0.23 ± 0.03 | 4960 ± 140 | 6.7 ± 0.7 | 0.66 ± 0.08 |

Kinetic parameters for recombinant His$_6$-tagged hsALAS2$_{\Delta N78}$ WT and XLP variant proteins expressed in *E. coli*. Reported parameters are best-fit values ±SE for progression curves (Supplementary Fig. 15) derived from the Prism software (GraphPad 8.0). Source data are provided as a Source Data file.

Ct-extension with the active site loop, providing an explanation for the increased rates of product release as detailed previously in stopped-flow experiments[9]. Our steady-state kinetic analysis of intact recombinant hsALAS2 WT and XLP variants (Supplementary Fig. 15b, c) reveals that the $k_{cat}$ and $V_{max}$ values for delAT and Q548X are 3- to 6-fold greater than WT and delG proteins (Table 3), in agreement with those reported in the literature[9,24] (delAGTG was not assayed due to C-terminal proteolysis and the putative formation of heterodimers). The association of the delG mutation with XLP has been hypothesized to be the result of increased in vivo enzyme stability relative to that of WT[24]. We show here that delG has the same 2- to 3-fold wild-type efficiency ($k_{cat}/K_m$) with respect to succinyl-CoA as delAT and Q548X (Table 3). Interestingly, this appears to be due in part to the lower $K_m$ of delG for succinyl-CoA compared to delAT, Q548X, and WT forms, suggesting that delG has a greater affinity for succinyl-CoA and this may contribute to its hyperactivity.

**Crystallography-based fragment screening**. We performed crystallography-based fragment screening to identify chemical ligands that serve as a starting point for development of small molecule therapy towards inherited defects of erythroid heme biosynthesis situated downstream of the ALAS2 reaction step, via a substrate reduction approach. Through solving 295 X-ray data sets (1.44–1.93 Å resolution) from hsALAS2$_{\Delta N142}$ crystals, each soaked with a different compound from a diverse fragment library, we identified 9 co-crystal structures each with a fragment bound to a hotspot region surrounding the Ct-extension (Fig. 5a and Supplementary Table 3 and Supplementary Data 1).

Fragments 1–8 (Fig. 5b) are accommodated in a pocket between helix α15 of one subunit and its interacting regions (helix α4) in the opposite subunit (Fig. 5c). This pocket is mainly hydrophobic in nature, contributed by Tyr574 from helix α15 and residues Phe267, Thr268, Lys271, Ile272, and Tyr413 from the opposite chain. Five of the fragments (1–5) extend into a subsite (referred to as 'Ia' in Fig. 5c) packing against Glu155 and Lys152 from the first helix (α1) of the catalytic core. As described above, helix α1 stabilizes the active site mobile loop and C-terminal helix α15, with contributions from Asp159 that forms the network of charged interactions among Asp159:Arg511:Glu569. The aromatic side-chain of Tyr574 (helix α15) adopts an alternate conformation when bound to fragments 2–4 (Supplementary Fig. 16). Three other fragments (6–8) occupy another subsite ('Ib' in Fig. 5c) in direct proximity to the last turn of helix α15 (e.g. Phe575 and Gly576).

A ninth fragment (9, Fig. 5b) was found at the opposite side of C-terminal helix α15 with respect to the above 8 fragments (Fig. 5c). This site is situated along the 2-fold axis of the hsALAS2 homodimer. One side of the fragment contacts a hydrophobic surface (Asn294, Ser295, Gly296, and Lys271) from its own subunit; its nitrile group captures hydrogen bonds with the main chain carbonyl of Phe575 (in C-terminal helix α15) and guanidinium side-chain from Arg293. The opposite face of the

fragment forms van der Waals contact with the same fragment bound to the dimeric subunit (9′).

We next interrogated the effects of the nine fragments in solution using our colorimetric activity assay. Activity of hsALAS2 was moderately inhibited by nearly 15% in the presence of 1 mM fragments 1 or 8 (Supplementary Fig. 17a), an effect that was not due to fragments alone interfering with the assay signal, and was also dependent on fragment concentration, reaching 28 and 22% inhibition with 5 mM fragments 1 and 8, respectively (Supplementary Table 4). MD simulations of *holo* hsALAS2 in the presence of fragment *1* further showed that the fragment remained bound to one protomer of the homodimer throughout the 50 ns interval and led to enhanced stabilization of R293-E571 salt bridge (Supplementary Fig. 17b), compared to fragment-free hsALAS2. Our data therefore suggest that hsALAS2 catalytic activity can be potentially modulated by small molecules that target the region close to the Ct-extension.

## Discussion

ALAS2 represents one of a handful of metabolic enzymes, where different mutations within the same protein can lead to distinct disorders associated with loss-of-function (enzyme deficiency) or gain-of-function (enzyme superactivity) disorders[40]. In particular, this study uncovers an autoinhibitory mechanism of ALAS2 mediated by its Ct-extension, providing the framework to explain how frameshift indel mutations in the ALAS2 C-terminus lead to the gain-of-function disorder XLP.

Previously, a conserved mobile loop in the rcALAS active site, undergoing conformational change to bind succinyl-CoA, was proposed as a regulatory feature that committed the enzyme to catalysis through an open-to-close active site transition[12]. Here, the hsALAS2 Ct-extension imposes additional conformational restriction on this active site loop (through the salt bridge network Glu569:Arg511:Asp159), preventing transition to the open conformation. To overcome these restraints, displacement of the Ct-extension from the conformation depicted in our structure, as supported by our MD simulations, is hence essential for both substrate binding and product release. Our data therefore add to the described complexity of regulatory mechanisms known to exist to modulate ALAS2 activity[41], underlining the importance of its tight regulation to maintain metabolic flux during erythropoiesis.

It is of note that others have shown that the Ct-extension of hsALAS2 most significantly impacts ALA release after succinyl-CoA binding[9], suggesting why product release is the rate-determining step. Previous studies[24] have highlighted that truncation of the Ct-extension does not significantly affect the $K_m$ for either substrate. Our data here, however, indicate that a lower $K_m$ for the delG XLP variant may contribute to its elevated catalytic efficiency. Transient kinetics monitoring the formation and decay of the intermediate quinoid species pointed to an increase in the rate of product release in the truncated forms of ALAS2 (ref. [9]). Our structure also provides insight into the role of engineered mutations of the active site loop from previous studies[33].

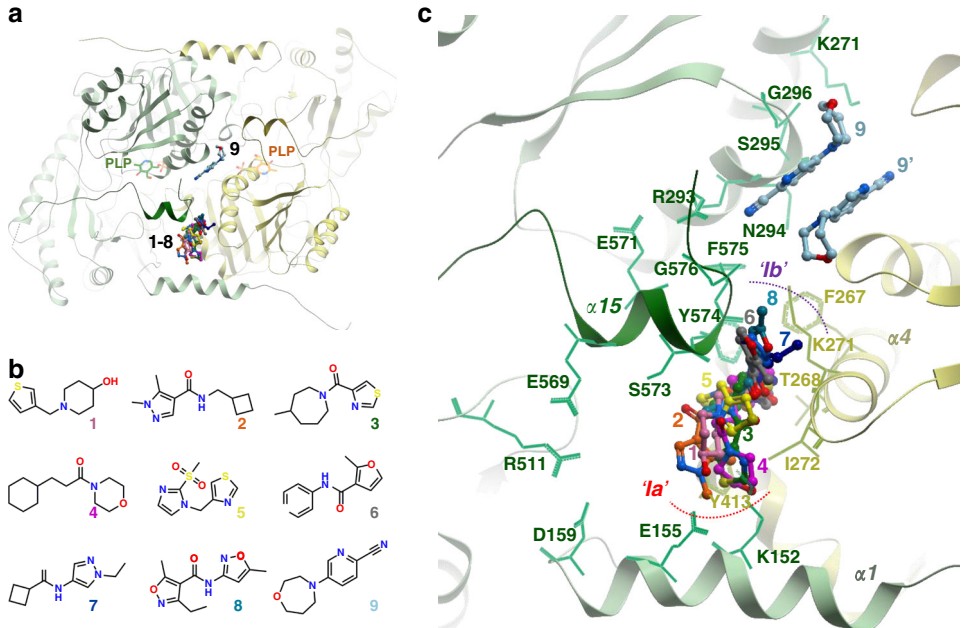

**Fig. 5 Crystallography-based fragment screening of hsALAS2. a** ALAS2-bound fragments are clustered into distinct regions of the protein. The two subunits of an ALAS2 homodimer are colored green and yellow. **b** Chemical structures of the fragments **1–9**, which are bound in proximity to the hsALAS2 Ct-extension. **c** Close-up view of the Ct-extension region of one subunit (green) and its neighboring subunit (yellow), showing the binding modes of fragments **1–9**. For fragment **9**, the second molecule (**9′**) bound to the yellow dimeric subunit is also shown.

Specifically, engineered mutations directed to the active site loop of murine ALAS2 also lead to a similar hyperactivity as the clinical mutations causing XLP. This observation can be easily reconciled with the hsALAS2 structure, as it would likely abolish interactions with the Ct-extension helix α15 and relieve the autoinhibition.

We speculate that the C-terminal regulatory feature, observed here for ALAS2, will also be present with a similar role in the more ubiquitous ALAS1 isozyme, considering their sequence conservation within this region (e.g. helix α15). However, beyond helix α15 at the far C-terminus, ALAS1 and ALAS2 have evolved very diverse amino acid sequences. Being surface-exposed, these residues may endow the two isoforms with different protein interaction functions. For example, the hsALAS2 Ct-extension is implicated from in vitro pull-down studies[24,36,37], as the binding site for the beta subunit SUCLA2 of the succinyl-CoA synthetase (SCS) complex in the TCA cycle[42]. It has been suggested that during erythropoiesis, the ALAS2-SUCLA2 interaction may play a role in directing the cellular pool of succinyl-CoA towards heme biosynthesis, away from its conversion to succinate in the TCA cycle[43]. Additionally, ALAS2 has been shown to be part of a larger heme synthesis metabolon whose role is currently undefined, but may involve coordinated regulation of ALAS2 and FECH, the last enzyme in the multi-step heme biosynthesis pathway[44]. Future studies may determine if and how these protein-protein interactions mediate the ALAS2 Ct-extension to regulate catalysis, most likely through the autoinhibitory mechanism shown in this study, but potentially also through facilitating the PLP loading process of the enzyme, shown to involve the AAA + protease ClpX[26].

Finally, our structural data provide a starting point for small molecule drug discovery. Three porphyria disorders can be classified as erythroid-specific, associated with loss of enzyme function in the heme biosynthetic pathway downstream of the ALAS2 step (Supplementary Fig. 18). These porphyrias result in the accumulation of toxic heme intermediates and point to the inhibition of ALAS2 as a potential therapeutic target via the substrate reduction approach that has precedence in other inherited metabolic disorders[45]. This is supported by the recent siRNA knockdown of the *ALAS1* mRNA showing effective prevention of adverse effects from acute hepatic porphyrias[46,47], pointing to a similar therapeutic benefit of ALAS2 inhibition for porphyrias in the erythropoietic pathway. Small molecule inhibitors that target the ALAS2 active site such as the itaconic acid derivative itaconyl-CoA[48], or take advantage of the enzyme's autoinhibitory mechanism mediated by the Ct-extension, are potential avenues for drug discovery. Relevant to this, fragments identified from our crystallography-based screening campaign have the potential for chemistry optimization into lead inhibitors in the future, aimed at increasing residence time at the Ct-extension region to prevent substrate access and product release during catalysis.

## Methods

**Protein expression and purification.** DNA fragments encoding hsALAS2 harboring different N- and C-terminal boundaries (Supplementary Fig. 2) were cloned into the pFB-LIC-Bse vector in-frame with a tobacco etch virus protease cleavable N-terminal His6-tag and expressed in baculovirus-infected Sf9 insect cells (Invitrogen) in Sf9 media. Site-directed mutagenesis (R511E) was carried out on hsALAS2$_{\Delta N142}$ and hsALAS2$_{\Delta N142 \Delta C545}$ cloned into pNIC28-Bsa4 vector using the QuikChange method, and confirmed by sequencing. In addition, hsALAS2$_{\Delta N78}$ containing NheI and BamHI restriction sites was cloned into the pET28a vector in-frame with the N-terminal His6-tag. Variants of the hsALAS2$_{\Delta N78}$:pET28a plasmid were prepared via Quickchange mutagenesis, including R511E, E569Q, and XLP mutants delAT, delAGTG, delG, and Q548X. All pET28a fusion proteins were transformed into BL21(DE3) competent cells and single-colony inoculations were induced at an OD600 of 0.5–0.7 with 0.1 mM IPTG for 16–18 h at 12 °C. Cultures were then centrifuged and cell pellets stored at −80 °C.

Proteins expressed in pFB-LIC-Bse and pNIC28-Bsa4 were lysed by sonication and centrifuged at 35,000×g. The clarified cell extract was incubated with 2.5 mL of Ni-NTA resin pre-equilibrated with lysis buffer (50 mM HEPES pH 7.5, 500 mM NaCl, 10 mM imidazole, 5% Glycerol, 0.5 mM TCEP). The column was washed with 100 mL Binding Buffer (50 mM HEPES pH 7.5, 500 mM NaCl, 5% glycerol, 10 mM imidazole, 0.5 mM TCEP), 50 mL wash buffer (50 mM HEPES pH 7.5, 500 mM NaCl, 5% glycerol, 40 mM imidazole, 0.5 mM TCEP) and eluted with 15 mL of Elution Buffer (50 mM HEPES pH 7.5, 500 mM NaCl, 5% glycerol, 250 mM imidazole, 0.5 mM TCEP). The eluant fractions were concentrated to 5 mL and applied to a Superdex 200 16/60 column pre-equilibrated in GF Buffer (50 mM HEPES pH 7.5, 500 mM NaCl, 0.5 mM TCEP, 5% glycerol). Eluted protein fractions were pooled and concentrated to 10–17 mg mL$^{-1}$. Where applicable, His6-tag removal was carried out by overnight treatment with TEV protease, followed by purification over Ni-NTA resin.

The hsALAS2 variants expressed in pET28a were similarly purified. Specifically, each protein was captured on a HisPur Cobalt Resin (Thermo Scientific) in solubilization buffer consisting of 50 mM Tris-HCl, pH 8.0, 20 μM PLP, 100 mM KCl, 10% glycerol, and 0.2% Tween-20. The resin was washed with solubilization buffer containing 15 mM imidazole. PMSF was also added at 0.5 mM during sonication, binding and washing steps. Elutions in 250 mM imidazole were buffer exchanged over Sephadex G-25 spin columns and quantitated with the Pierce BCA Protein Assay Kit (Thermo Scientific). Purified protein samples (1 μg for SDS-PAGE and 100 ng for western blots) were run out on Mini-PROTEAN Stain-Free polyacrylamide gels (Bio-Rad) and bands detected using the ChemiDoc MP imaging system (Bio-Rad) and ImageLab software (version 5.1, Bio-Rad). Multiplex western blotting was carried out by transferring gel bands to low-fluorescence PVDF membrane, followed by blocking the membrane in 2% BSA at room temperature for one hour and incubation overnight at 4 °C in 1:10,000 rabbit whole-serum polyclonal hsALAS2 (custom-made, purified antibody from Invitrogen available through Thermo Scientific, catalog number 21553) and 1:5000 mouse monoclonal anti-6xHis (Proteintech, catalog number 66005-1-lg) primary antibodies in 1% BSA. After washing extensively in TBST, the membrane was incubated in 1:50,000 dilutions of Invitrogen fluorescent donkey anti-rabbit IgG (DyLight 550, catalog number SA5-10039) and goat anti-mouse IgG (DyLight 650, catalog number SA5-10041) secondary antibodies (Thermo Scientific) before washing again and imaging on the ChemiDoc system. Uncropped and unprocessed scans of blots and gels are provided in the Source Data file.

**Crystallization and structure determination.** Crystals were grown by the vapor diffusion method. To crystallize hsALAS2$_{\Delta N142}$, protein was pre-incubated with 5 mM hydroxylamine for 30 min on ice to convert PLP into the non-covalent form. Sitting drops containing 75 nL protein (17 mg mL$^{-1}$) and 75 nL well solution containing 25% (w/v) PEG 3350, 0.1 M Bis-Tris pH 6.7 and 0.3 M MgCl$_2$ were equilibrated at 20 °C. To crystallize hsALAS2$_{\Delta N78}$, sitting drops containing 200 nL protein (11 mg mL$^{-1}$), 133 nL well solution containing 20% (w/v) PEG3350 and 0.2 M NH$_4$Cl, and 67 nL seed solution derived from previously obtained micro crystals. Crystals were cryo-protected using 25% (v/v) ethylene glycol and flash-cooled in liquid nitrogen. Diffraction data were collected at the DLS beamline I04 and I24.

HsALAS2$_{\Delta N142}$ was crystallized in the monoclinic space group C2 with two molecules in the asymmetric unit. The data was processed using the Xia2 autoprocessing dials pipeline.

hsALAS2$_{\Delta N78}$ was crystallized in the monoclinic space group P2$_1$ with four molecules in the asymmetric unit. The data were processed using non-ellipsoidal anisotropy truncations by the STARANISO server, based on a lower limit 1.2 for the local I/sigI producing resolution cutoff of 2.65 Å in the best direction and 3.55 Å in worst. The human structures were solved by molecular replacement using the program PHASER from CCP4 (ref. [49]) and the rcALAS structure (PDB 2BWN)[12] as the search model. The final model was produced by iterative cycles of restrained refinement and model building using COOT[50], REFMAC5 from CCP4 (ref. [51]) and Phenix.refine[52].

The final model consists of Phe143-Met578 of chains A and B. In both chains the last 9 residues, aa 579–587, and a loop between Ser182 and Ser188 were not visible in the electron density map indicating that this region was largely unstructured. Additionally, chain-to-chain variation in electron density quality and loop orientation were found between residues Asp549 and Cys558, likely reflective of differing crystal packing restraints and an intrinsic tendency for conformational heterogeneity within these regions. Coordinates and structure factors for hsALAS2$_{\Delta N142}$ were deposited in the PDB with the accession code 6HRH. Statistics for data collection and refinement are summarized in Table 1.

**Crystallography-based fragment screening.** To grow crystals for the fragment screening campaign, 10 mg mL$^{-1}$ of hsALAS2$_{\Delta N142}$ was pre-incubated with 5 mM hydroxylamine for 30 min on ice to convert PLP into the homogenous, non-covalent form. Crystals were grown by vapor diffusion in 400 nL sitting drops in the presence of seeds at 20 °C equilibrated against well solutions of 0.1 M Bis-Tris pH 7.0, 0.3 M magnesium chloride and 23% PEG3350. For soaking, 50 nL of each fragment compound (final concentration of 125 mM) was added to a crystallization drop using an ECHO acoustic liquid handler dispenser at the Diamond light source XChem facility. Crystals were soaked for two hours with fragments from the DSi-Poised Library (Diamond Light Source Ltd) before being harvested using XChem SHIFTER technology, cryo-cooled in liquid nitrogen, and data sets collected at the beamline I04-1 in "automated unattended" mode. The XChemXplorer pipeline[53] was used for structure solution with parallel molecular replacement with DIMPLE from CCP4 (ref. [54]), followed by map averaging and statistical modeling to identify weak electron densities generated from low occupancy fragments using PANDDA software[55]. Coordinates and structure factors for data sets with the 9 bound fragments have been deposited in the PDB (Supplementary Table 3), and refinement statistics are found in Supplementary Data 1.

**SAXS.** SAXS experiments for the hsALAS2$_{\Delta N142}$ and hsALAS2$_{\Delta N142\Delta C545}$ were performed at 0.99 Å wavelength on the Diamond Light Source beamline B21 coupled to the Shodex KW403-4F size exclusion column (Harwell, UK) and

equipped with Pilatus 2 M two-dimensional detector at 4.014 m distance from the sample, $0.005 < q < 0.4$ Å-1 ($q = 4\pi \sin \theta/\lambda$, $2\theta$ is the scattering angle). The samples were prepared in a buffer containing 300 mM NaCl, 25 mM Hepes 7.5, 1 mM TCEP, 2% glycerol, 1% sucrose, and the measurements were performed at 20 °C. The data were processed and analyzed with Scatter[56] and the ATSAS program package[57]. Scatter was used to calculate the radius of gyration Rg and forward scattering I(0) via Guinier approximation, and to derive the maximum particle dimension Dmax and P(r) function. The ab initio model was derived using DAMMIF[57]. Ten individual models were created, then overlaid and averaged using DAMAVER[57].

**Activity assays and kinetics.** To measure enzyme activity of hsALAS2$_{\Delta N78}$ and hsALAS2$_{\Delta N142}$ proteins, discontinuous colorimetric activity assays were conducted per Shoolingin-Jordan et al.[58] with modifications. Reaction mixtures consisted of 50 mM potassium phosphate buffer, pH 7.4, 50 mM PLP, 1 mM DTT, 10 mM MgCl$_2$, various concentrations of glycine and succinyl-CoA (Sigma), and 1–4 μg mL$^{-1}$ freshly purified hsALAS2 enzyme (175 μL total). After incubation at 37 °C for 15 min (previously checked for linear ALA formation with each enzyme concentration), reactions were terminated with 100 μL trichloroacetic acid and centrifuged at 13,000×g for 5 min to remove protein. Supernatants (240 μL) were added to freshly prepared mixtures of 240 μL of 1 M sodium acetate, pH 4.7, and 20 μL acetylacetone (500 μL total), and boiled for 10 min to derivatize the ALA product. Samples were cooled and three 150 μL aliquots per reaction (two or three technical replicates) were further derivatized with 150 μL modified Ehrlich's reagent and monitored at 554 nm every 60 s in a CLARIOstar microplate reader (BMG Labtech). Assays containing small-molecule fragments were scaled down to 140 μL reaction mixtures (two technical replicates) and included 5% v/v (7 μL) of fragment stock solutions or DMSO diluent. Absorbance values collected after 5 min were converted to molar quantities of ALA using an extinction coefficient of 60.4 mM$^{-1}$ cm$^{-1}$. For kinetic studies, apparent $V_{max}$, $K_m$, and $k_{cat}$ values were determined by titrating 2.5–50 mM glycine in the presence of 100 μM succinyl-CoA and 5–100 μM succinyl-CoA in the presence of 50 mM glycine. Michaelis–Menten nonlinear regression analysis was subsequently carried out on data for two or three separate protein preparations (two or three biological replicates) with Prism software (GraphPad 8.0). As control, we confirmed that the His$_6$-tag on the recombinant protein did not influence hsALAS2 activity, as shown for WT and R511E variant of hsALAS2$_{\Delta N142}$ (Fig. 3d, e and Table 2).

**Differential scanning fluorimetry.** HsALAS2$_{\Delta N142}$ was assayed for shifts in melting temperature caused by titration of PLP and succinyl-CoA in a 96-well PCR plate using an Mx3005p RT-PCR machine (Stratagene). Each reaction well (20 μl) consisted of protein (20 μM in a buffer containing 50 mM Hepes, pH 7.5, 500 mM NaCl), SYPRO-Orange (Invitrogen) diluted 1000×. Fluorescence intensities were measured from 25 °C to 96 °C with a ramp rate of 1 °C min$^{-1}$, as described[59]. The temperature shifts, $\Delta T_{m,obs}$, for each ligand were determined. The experiment was repeated three times and an error was calculated based on the standard deviation.

**Flourescence intensity.** HsALAS2$_{\Delta N142}$ was assayed for shifts in fluorescence intensity caused by titration of PLP. Each well (10 μl) consisted of protein (20 μM in a buffer containing 50 mM Hepes, pH 7.5, 500 mM NaCl) and varying concentrations of PLP. Schiff base formation was monitored via excitation at 440 nm and emission at 520 nm using an Omega plate reader. The experiment was repeated three times and an error was calculated based on the standard deviation.

**Modeling hsALAS2 for MD simulation.** The crystal structure of hsALAS2 (this study) was used for all MD simulation studies. Missing residues were modeled into the crystal structure based on a homology model of hsALAS2 constructed with the scALAS crystal structure (PDB 5TXT) as a template using the MODELLER program[60]. The homology model was first structurally aligned with the hsALAS2 crystal structure, and the coordinates of the missing residues were then taken from the homology model and incorporated into the crystal structure. The structure generated, after an energy minimization step, thereby included residues 137–578 from both monomeric units. Three different systems were modeled: (i) hsALAS2 apo form (without PLP), (ii) hsALAS2 holo form with PLP covalently attached to Lys391 (but without substrates), and (iii) hsALAS2 holo form with substrates glycine-PLP and succinyl-CoA.

In order to model system (iii), the starting structure for the protein was chosen from the simulation on system (ii) by clustering all conformations sampled in the trajectory and then choosing the structure at the center of the largest cluster as the representative structure. PLP was then removed from Lys391 in the representative structure. The substrates glycine-PLP and succinyl-CoA were modeled into this structure by using the substrate-bound *Rhodobacter capsulatus* ALAS (rcALAS) structure as a reference template. The structure with PDB ID 2BWP (rcALAS with glycine-PLP) was first aligned with the representative hsALAS2 structure, and the resulting orientation of glycine-PLP was modeled into the representative hsALAS2 structure. In a similar manner, the structure with PDB ID 2BWO (rcALAS with succinyl-CoA) was used for modeling succinyl-CoA into the representative hsALAS2 structure. Both the monomeric units of the representative structure included the two substrates. In order to remove possible bad contacts

between the substrates and the protein, minimization and equilibration of the system was performed in the presence of restraints on the heavy atoms of the protein (a restraining force with force constant $1000 \, kJ \, mol^{-1} \, nm^{-2}$ was used).

All MD simulations were performed in the NPT ensemble (constant number of particles (N), constant pressure (P), and constant temperature (T)) for 50 ns using the GROMACS program with the CHARMM36 all-atom force field[61,62] and the TIP3P water model[63] (Supplementary Software 1). A time-step of 2 fs was used, and all covalent bonds involving hydrogen atoms were constrained using the LINCS algorithm[64]. A cut-off of 12 Å was used for short-range non-bonded interactions, and the particle mesh Ewald method was employed for long-range electrostatics[65]. Force field parameters for PLP-lysine, PLP-glycine, and succinyl-CoA were obtained from the Paramchem server[66,67]. Details of the simulation box for all systems are shown in Supplementary Table 5. Two sets of simulations were performed for all the systems.

To study the conformational dynamics of ALAS2 R511E variant, the crystal structure of hsALAS2$_{\Delta N142}$ was used to perform the R511E substitution using the CHARMM-GUI server[68]. The simulation protocol was the same as that followed for wild-type ALAS2.

**Reporting summary**. Further information on research design is available in the Nature Research Reporting Summary linked to this article.

## Data availability
All coordinates and structure factors were deposited to the Protein Data Bank under the accession codes PDB 6HRH [https://doi.org/10.2210/pdb6HRH/pdb], PDB 5QQY [https://doi.org/10.2210/pdb5QQY/pdb], PDB 5QR1 [https://doi.org/10.2210/pdb5QR1/pdb], PDB 5QRA [https://doi.org/10.2210/pdb5QRA/pdb], PDB 5QRC [https://doi.org/10.2210/pdb5QRC/pdb], PDB 5QRD [https://doi.org/10.2210/pdb5QRD/pdb], PDB 5QQW [https://doi.org/10.2210/pdb5QQW/pdb], PDB 5QQX [https://doi.org/10.2210/pdb5QQX/pdb], PDB 5QRE [https://doi.org/10.2210/pdb5QRE/pdb], PDB 5QQU [https://doi.org/10.2210/pdb5QQU/pdb]. Main data supporting the findings of this study are available within the article and the Supplementary Information. Raw data for Tables 2 and 3, Supplementary Table 4, Fig. 3b–e, and Supplementary Figs. 3a–c, 4a–c, 5, 6, 9–12, 13a, 15a–c, and 17a, b are available in the Source Data file. Other data are available from the corresponding author upon reasonable request.

## Code availability
GROMACS codes for MD simulations reported in Supplementary Figs. 9–11, 12b, 13a, and 17b are provided as Supplementary Software 1.

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

## Acknowledgements

The Structural Genomics Consortium is a registered charity (Number 1097737) that receives funds from AbbVie, Bayer Pharma AG, Boehringer Ingelheim, Canada Foundation for Innovation, Eshelman Institute for Innovation, Genome Canada, Innovative Medicines Initiative (EU/EFPIA) [ULTRA-DD grant no. 115766], Janssen, Merck & Co., Novartis Pharma AG, Ontario Ministry of Economic Development and Innovation, Pfizer, São Paulo Research Foundation-FAPESP, Takeda, and Wellcome Trust [092809/Z/10/Z]. H.J.B. is further supported by the Nuffield Department of Clinical Medicine and Medical Research Council. R.J.D. and D.F.B. were supported in part by the Department of Genetics and Genomic Sciences, Icahn School of Medicine at Mount Sinai.

## Author contributions

G.A.B., H.A.D., and W.W.Y. designed the experiments. H.J.B., G.A.B., J.R.M., S.P., W.R.F., E.R., A.R., and D.F.B. performed the experiments. H.J.B., J.R.M., R.J.D., G.B., H.A.D., and W.W.Y. wrote the manuscript.

## Competing interests

The authors declare no competing interests.
