## [Peer Review File · Nature Communications]

Reviewers' comments:

Reviewer #1 (Remarks to the Author):

I am a computational chemist by training and have focused in my evaluation on the computational aspects. Nonetheless, I have some questions to the experimental parts as well.

The manuscript describes the determination of a structure of an N-terminally truncated version of the human aminolevulinate synthase (hsALAS2). The C-terminal extension (Ct-extension) was found to be in a position where it has an autoinhibitory effect, which offers new explanations for observed genetic disorders. The authors study the dynamics of the Ct-extension using molecular dynamics simulations. Furthermore, they have performed kinetic studies on some mutants and performed a crystallographic screen for putative inhibitors. The work seems carefully done, is clearly described, describes novel findings and addresses a relevant topic.

I would like to make the following comments:

1. The authors perform simulations of three distinct states of the protein (and one fourth, with a putative inhibitor bound). In order to model the substrate bound state, they start from a representative structure of the most common cluster of the simulation with PLP bound, but without substrate. This means that in that simulation, the Ct-extension already moves away from the position that is observed in the X-ray structure. That is a discrepancy that needs to be discussed. Seemingly, the dynamics in the Ct-extension already take place when the substrates are not bound?
2. The authors have modeled missing residues in the structure based on a homology model that was based on the scALAS structure. Does this also include the loop 549-555, which was not observed in the hsALAS2 structure? That would not make sense as figure 2c,d and the text in lines 210 – 212 shows that these structures are very different in this region. As this flexible loop is very likely to affect the dynamics of the Ct-extension, the authors should describe in more detail how it was modeled. Could an inappropriate model of this loop lead to enhanced dynamics of the entire Ct-extension and thereby lead to the movement of helix 15, already in the substrate free structure? See point 1.
3. It is difficult to assess if the asymmetry between the monomers, which is described in the lower panels of supplementary figures 10 and 11 is a true observation, or if this is just luck. As the systems seem much more dynamic, two seemingly random trajectories may seem asymmetric. This can only be resolved by performing technical replicates of the simulations.
4. The PLP in supplemental figure 7 seems different from the structure in figure 1e. The carbonyl oxygen is colored blue, and points in a very different direction in figure S7.

5. Relevant details on the methodology of the molecular simulations are missing. How big was the simulation box, how were long-range electrostatics included, was the simulation performed at constant temperature and/or pressure, were bond-lengths constrained, what timestep is used? The Gromacs code is not listed under 'software' in the 'reporting summary'.

6. The identified fragments are tested for their inhibitory capacity by a single point experiment. For an inhibitor of this kind, it may be expected that full inhibition is not attainable. Determination of an IC50 would be appropriate to confirm a concentration dependent inhibition.

7. More importantly, the inhibition of the WT is not the relevant measure here. Would the authors not much rather show inhibitory behavior of these compounds against delAT or Q548X ?

8. Abbreviations ClpX and AAA+ are not explained.

9. Homology is not quantifiable. Either two proteins are homologous (have a common ancestor) or they are not. In line 83, the authors describe a low-homology region. I guess they mean a region with low sequence similarity.

10. Very few typo's

a. line 66: aldmine -> aldimine

b. line 184: through pi-pi -> through pi-pi interactions

c. line 319: two opening brackets

Reviewer #2 (Remarks to the Author):

This is an interesting paper that gives a bit more detail on the inhibitory action of the C-terminal region of aminolevulinic acid synthase. Alas2 has been recently shown to be a good target to control certain porphyrins through gene silencing approaches so there is potential therefore for small molecule drugs.

The authors do repeat a lot of what is known already, but the paper does provide a structural interpretation of the C-terminal autoinhibition. The MD is pretty, but I am not sure it is that especially informative in this case.

1. The key crystallographic point addressed here is the conformation of the C-terminal region because the story revolves around the structure of this autoinhibitory loop. I would therefore expect

to see an omit map for this region to confirm its architecture and its correct placement in the electron density.

2. I suspect the problem the authors have is that most of the C-terminal region is unstructured, flexible, or disordered. At least some of the C-terminal region should be well-structured to warrant the crystallographic build-up in the narrative.

Provided there is good evidence for the C-terminal structure, then Nature Communications seems about the right level for this paper.

Reviewer #3 (Remarks to the Author):

A combination of crystallography, limited biochemical analysis of disease-causing mutations, and crystallography-based fragment screening were employed in this study on human ALAS2. MD simulations are included to support mechanistic inferences. The significance of the work is that the structural studies led to mapping of the C-terminal extension, providing a framework for formulating hypotheses about the molecular basis of disease-causing mutations for XLSA and XLP. Overall, the experiments seem to have been well conducted. However, the mechanistic insights to emerge from the study are limited. The fragment search led to the identification of ligands that were largely clustered in a hydrophobic pocket but had minimal inhibitory effects (~15% at 1 mM concentration). A key conclusion that the R511E mutation destabilizes the structure of the Ct-extension was not experimentally supported and other inferences of gain-of-function XLP mutation, tended to be overstated. The absence of quantitative data (kinetic data are shown without error bars, the number of replicates are not mentioned, values for kinetic parameters are not provided for wild-type versus mutant ALAS2) diminishes the study's rigor.

Major

1. Line 126. The K_m values for glycine and succinyl-CoA are not reported as stated in Fig S4a,b. These need to be noted with SDs in the text.

2. Line 129. The data shown in Fig 4c does not establish interaction between ALAS2 and SUCLG1-SUCLA2, but rather a messy gel with many bands. If the authors consider this conclusion regarding protein-protein interaction to be important for this study, then MS data establishing the identity of the proteins present in the putative crosslinked bands needs to be shown.
3. Line 261/2: The statement that breaking the R511:E569 interaction renders the Ct-extension highly destabilized is not backed by experimental evidence. Furthermore, it is only marginally prone to proteolysis. Hence the conclusion that the R511 mutation is essential for maintaining the structural integrity of the Ct-extension is overstated.
4. His-tags were retained during characterization of mutant proteins, which is problematic. The authors need to remove the Hi-tags to make meaningful inferences about the resulting disease phenotype.
5. Line 272. The conclusion that the elongated C-terminal extension in the delAGTG variant is subjected to proteolysis is not experimentally supported. While a small amount of proteolysis is seen, its location is not mapped to the Ct-extension.
6. Fig 4c,d. Error bars are not shown for the kinetic data. The number of replicates is not mentioned.
7. Lines 279-286. The values of k_{cat} and K_m for wild-type versus each mutant that was characterized, need to be reported together with S.D.
8. Line 283. Although it is stated that “we show here that delG has the same 2- to 3-fold wild-type efficiency (k_{cat}/K_m) with respect to succinyl-CoA as delAT and Q548X”, no such data are provided.
9. Line 335. The statement, “a catalytically grounded state that prevents transition to the open conformation” is unclear. What is meant by catalytically grounded? And how can the authors rule out the much more like ensemble of conformational substates, with the equilibria being shifted by individual mutations?
10. Line 342. What is the experimental basis of the conclusion that kinetically, “the Ct-extension exerts more influence towards product release rather than succinyl-CoA binding?” K_m can not be equated with substrate affinity.

Minor:

1. Line 29. Change “That has evolved among higher eukaryotes” to “that is only present in higher eukaryotes”.
2. Fig 1b. Monomer A described as being light green in the figure legend is shown in grey in the figure. Same problem in Fig. 1d, subdomain 2 and Fig 1e.
3. Lines 90/91 (Supplemental section). Change “that could accommodate for the disordered” to “that could account for the disordered”.
4. Line 184. Change “ π - π ” to “ π - π interactions”.

5. Line 191. FigS1 does not show the structure of succinyl-CoA bound ALAS2. Please correct.

6. Line 365. Define FECH.

Reviewer #1 (Remarks to the Author):

1. The authors perform simulations of three distinct states of the protein (and one fourth, with a putative inhibitor bound). In order to model the substrate bound state, they start from a representative structure of the most common cluster of the simulation with PLP bound, but without substrate. This means that in that simulation, the Ct-extension already moves away from the position that is observed in the X-ray structure. That is a discrepancy that needs to be discussed. Seemingly, the dynamics in the Ct-extension already take place when the substrates are not bound?

Response: As illustrated in the figure below, the orientation of the Ct-extension in the PLP-bound form (without substrate) is very similar to that in the crystal structure. The Ct-extension only moves away from its position in the crystal structure, after binding of the succinyl-CoA substrate (Fig. 3), as shown by complete breakage of the R511:E569 and R293/K299:E571 salt bridges in our simulation (Supplementary Figs. 9-11).

Orientation of the Ct-extension. The Ct-extension from monomer A in the energy minimized crystal structure (green) and in the representative structure from the simulation for PLP-bound ALAS2 (magenta).

2. The authors have modeled missing residues in the structure based on a homology model that was based on the scALAS structure. Does this also include the loop 549-555, which was not observed in the hsALAS2 structure? That would not make sense as figure 2c,d and the text in lines 210 – 212 shows that these structures are very different in this region. As this flexible loop is very likely to affect the dynamics of the Ct-extension, the authors should describe in more detail how it was modeled. Could an inappropriate model of this loop lead to enhanced dynamics of the entire Ct-extension and thereby lead to the movement of helix 15, already in the substrate free structure? See point 1.

Response: It is correct that a few residues were missing in the crystal structure, namely 182-187 and 549-555 in chain A, and 183-187 and 549-557 in chain B. To model a complete hsALAS2 structure (137-578) with missing residues included, we first constructed a hsALAS2 homology model using the scALAS2 structure (PDB ID: 5TXT) as a template. This hsALAS2 homology model was then superimposed with the hsALAS2 crystal structure, such that coordinates for the missing residues

were extracted from the homology model, and covalently linked to the crystal structure. A energy minimization step was then performed. The above procedures have now been included in the Materials and Methods section, page 18.

The figure below shows the region 549-555 that was extracted from the homology model. We would like to make it clear that this small stretch of 9 amino acids connects the last helix of the catalytic domain to the Ct-extension, and is not involved in shielding the active site. The orientation of the Ct-extension in our model is based solely on the crystal structure, and hence very unlikely to be biased by the small stretch taken from the homology model (please also see the figure in response to comment 1). The enhanced dynamics of helix 15 is seen only when bound with succinyl-CoA substrate in the simulation, as described in the response to the previous comment.

Modeling missing residues. The missing residues (residues 549-555) and the Ct-extension from monomer A of hsALAS2. For modeling the missing residues, a homology model of hsALAS2 was first constructed using scALAS2 (PDB ID: 5TXT) as a template. This homology model was structurally aligned with the crystal structure of hsALAS2, and the coordinates of the missing residues were extracted from the model. The extracted residues were then covalently linked to the crystal structure, and the structure was energy minimized.

3. It is difficult to assess if the asymmetry between the monomers, which is described in the lower panels of supplementary figures 10 and 11 is a true observation, or if this is just luck. As the systems seem much more dynamic, two seemingly random trajectories may seem asymmetric. This can only be resolved by performing technical replicates of the simulations.

Response: We have now performed one replicate simulation for each of the three systems, by assigning a different set of initial velocities to all the atoms in the system. As shown in the figure below, the asymmetric behaviour in the salt bridge dynamics of the two monomeric subunits in the presence of substrates is observed in the replicate simulation; this is in agreement with the original simulation reported in the manuscript. We have added a sentence in the legend of Supplementary Fig. 10 and 11 to indicate this is n=2 experiment.

Dynamics of salt bridges involving E569. The variation of the distance between the residues participating in salt bridges is shown as a function of simulation time. To calculate the distance between two residues, the center of a given residue was defined as the center of mass of the side-chain guanidinium/amine/carboxylate group, depending on the amino acid type. The distance between the residue centers was then calculated. While R511-E569 is a native salt bridge present in the crystal structure, E569-R572 is a non-native salt bridge.

Dynamics of salt bridges involving E571. The variation of the distance between the residues participating in salt bridges is shown as a function of simulation time. To calculate the distance between two residues, the center of a given residue was defined as the center of mass of the side-chain guanidinium/amine/carboxylate group, depending on the amino acid type. The distance between residue centers was then calculated. While

R293-E571 and K299-E571 are native salt bridges present in the crystal structure, E571-R572 is a non-native salt bridge.

4. The PLP in supplemental figure 7 seems different from the structure in figure 1e. The carbonyl oxygen is coloured blue, and points in a very different direction in figure S7.

Response: We have now provided a corrected version of Supplementary Fig. 7.

5. *Relevant details on the methodology of the molecular simulations are missing. How big was the simulation box, how were long-range electrostatics included, was the simulation performed at constant temperature and/or pressure, were bond-lengths constrained, what timestep is used? The Gromacs code is not listed under 'software' in the 'reporting summary'.*

Response: Details of the simulation methodology are now included in the Materials and Methods section. Simulation box details are given in a new Supplementary Table 5. GROMACS codes have been included under 'Software' in the Reporting Summary.

6. *The identified fragments are tested for their inhibitory capacity by a single point experiment. For an inhibitor of this kind, it may be expected that full inhibition is not attainable. Determination of an IC50 would be appropriate to confirm a concentration dependent inhibition.*

Response: The fragment hits identified from our crystallography screening are low-molecular-weight compounds (average < 300 Da). In our experience from several fragment screening campaigns, these initial fragment hits are not expected to exhibit much inhibitory potency. We therefore thank the reviewer for understanding that full inhibition would not be attainable, and as such there would not be much meaning to calculate IC50 (half-maximal inhibition). However, as a follow-up to our original inhibition studies of wild-type hsALAS2 activity in the presence of 1 mM fragment concentrations, we have now determined wild-type enzyme activity in a separate set of experiments using 1 mM and 5 mM concentrations of fragments 1 and 8 (fragments that showed significant inhibition in the original activity assays at 1 mM) and have observed concentration-dependent inhibition. These data are incorporated into Supplementary Table 4, and cited in the text on page 11.

7. *More importantly, the inhibition of the WT is not the relevant measure here. Would the authors not much rather show inhibitory behavior of these compounds against delAT or Q548X?*

Response: We evaluated the inhibitory effect of fragments using wild-type protein, but not delAT or Q548X, for two reasons:

- The fragment hits of interest to us (1-8) are clustered around the Ct-extension; this region would not be present in the truncated delAT and Q548X variants.
- The primary motivation for our fragment-based discovery is to provide inhibitor starting point as substrate reduction therapy for those erythroid porphyria disorders caused by enzyme defects downstream of ALAS2 (namely UROS, UROD, FECH). It was outside the scope of our current fragment campaign to target the gain-of-function erythroid protoporphyria due to delAT or Q548X; if it were, we should be using a construct more relevant to these variant proteins, such as the catalytic domain-alone protein, which we currently have not crystallised.

We have now clarified the above points in the discussion section.

8. *Abbreviations ClpX and AAA+ are not explained.*

Response: Full names have now been added in their first mention, on page 4.

9. Homology is not quantifiable. Either two proteins are homologous (have a common ancestor) or they are not. In line 83, the authors describe a low-homology region. I guess they mean a region with low sequence similarity.

Response: We have now replaced 'low-homology' with 'poor sequence conservation' on page 4.

10. Very few typo's

a. line 66: *alldmine* -> *aldimine*

b. line 184: *through pi-pi* -> *through pi-pi interactions*

c. line 319: *two opening brackets*

Response: Typos have now been corrected.

Reviewer #2 (Remarks to the Author):

1. *The key crystallographic point addressed here is the conformation of the C-terminal region because the story revolves around the structure of this autoinhibitory loop. I would therefore expect to see an omit map for this region to confirm its architecture and its correct placement in the electron density.*

Response: We have now included an omit map for the Ct-extension region, incorporated into Supplementary Fig 8, to confirm its topology and correct placement in the electron density.

2. *I suspect the problem the authors have is that most of the C-terminal region is unstructured, flexible, or disordered. At least some of the C-terminal region should be well-structured to warrant the crystallographic build-up in the narrative.*

Response: As shown in the omit map described above (Supplementary Fig. 8a), the majority of the Ct-extension is ordered and can be modelled from well-defined electron density. The only disordered segments in the vicinity are residues 549-557 (refer to reviewer 1) which connect the catalytic domain to the beginning of the Ct-extension, as well as the last 7 residues at the C-terminus. Both segments are not expected to play a role in shielding and gating the enzyme's active site.

Reviewer #3 (Remarks to the Author):

The absence of quantitative data (kinetic data are shown without error bars, the number of replicates are not mentioned, values for kinetic parameters are not provided for wild-type versus mutant ALAS2) diminishes the study's rigor.

Response: We have now corrected all these in the relevant figures and text reporting enzyme kinetics data.

Major

1. Line 126. The Km values for glycine and succinyl-CoA are not reported as stated in Fig S4a,b. These need to be noted with SDs in the text.

Response: The Km values have been added as insets to the progression curves in Supplementary Fig. 4a,b, and also to the text on page 5.

2. Line 129. *The data shown in Fig 4c does not establish interaction between ALAS2 and SUCLG1-SUCLA2, but rather a messy gel with many bands. If the authors consider this conclusion regarding*

protein-protein interaction to be important for this study, then MS data establishing the identity of the proteins present in the putative crosslinked bands needs to be shown.

Response: We thank reviewer 3 for this suggestion. We have now provided, as addendum to Supplementary Fig. 4c, MS/MS analysis confirming the identity of ALAS2, SUCLG1 and SUCLA1 peptides in the relevant SDS-PAGE bands from the crosslinking experiment.

3. Line 261/2: The statement that breaking the R511:E569 interaction renders the Ct-extension highly destabilized is not backed by experimental evidence. Furthermore, it is only marginally prone to proteolysis. Hence the conclusion that the R511 mutation is essential for maintaining the structural integrity of the Ct-extension is overstated.

Response: We agree that the SDS-PAGE and western blot data (Fig. 4b) for the E569Q variant indicate only marginal C-terminal proteolysis, but emphasize here that the R511E variant is also proteolyzed to a more significant extent than E569Q. Because the R511 active site loop residue interacts with both E569 and E571 residues in the C-terminal α 15 helix (Fig. 2a), one would expect the R511E substitution to completely disrupt this interaction (making the C-terminus highly susceptible to proteolysis) while the R511:E571 interaction may still be conserved in the E569Q variant (thus making it less susceptible proteolysis compared to the R511E variant).

4. His-tags were retained during characterization of mutant proteins, which is problematic. The authors need to remove the Hi-tags to make meaningful inferences about the resulting disease phenotype.

Response: We reason that the His-tagged region does not pose a problem in our inference of variant proteins from enzyme assay, because (i) the His-tag is surface-exposed and distant from the enzyme catalytic site, as revealed by our crystal structure; (ii) all wild-type and variant proteins have the same His-tag configuration hence data generated from them are entirely comparable.

5. Line 272. The conclusion that the elongated C-terminal extension in the delAGTG variant is subjected to proteolysis is not experimentally supported. While a small amount of proteolysis is seen, its location is not mapped to the Ct-extension.

Response: The SDS-PAGE and western blot (Fig. 4b) both show two bands (higher MW band much more intense than the lower), indicating mild but clear proteolysis of the recombinant delAGTG enzyme. More importantly, antibody detections of both the 6xHis N-terminal tag and ALAS2 catalytic domain also yield two bands, inferring that proteolysis is localized to the C-terminal region. (Note that we mistakenly omitted a description of the western analysis in the Fig. 4b legend but that this information has now been added.) While we have not conducted MS analysis of the delAGTG protein presented here to identify the specific point of cleavage, recombinant delAGTG expressed in *E. coli* has been previously shown by a co-author on this manuscript to be susceptible to C-terminal proteolysis by SDS-PAGE and MS analysis (see ref. 24).

6. Fig 4c,d. Error bars are not shown for the kinetic data. The number of replicates is not mentioned.

Response: In lieu of error bars, all individual points have been plotted to clearly indicate experimental variation, in the new Fig. 4c and d. Standard error values from the nonlinear regression analysis have been reported in Table 2, and this reference to Table 2 has been added to the Fig. 4 legend. In addition to the two biological replicates (assays of two separate protein preps) in the first manuscript submission, a third replicate experiment has been conducted to further support the original data (the Methods section has been edited to reflect this change). Points for

each of the three biological replicate titrations (N=3) are indicated with distinct symbols (circles, boxes, and triangles) and represent the average of three technical replicates.

7. Lines 279-286. The values of k_{cat} and K_m for wild-type versus each mutant that was characterized, need to be reported together with S.D.

Response: All the k_{cat} and K_m data are found in Table 2, which we missed out referencing in the relevant sentence on page 10. We have now cited Table 2 accordingly.

8. Line 283. Although it is stated that “we show here that delG has the same 2- to 3-fold wild-type efficiency (k_{cat}/K_m) with respect to succinyl-CoA as delAT and Q548X”, no such data are provided.

Response: Again, the k_{cat}/K_m data are found in Table 2, which we missed out referencing in the relevant sentence on page 10. We have now cited Table 2 accordingly.

9. Line 335. The statement, “a catalytically grounded state that prevents transition to the open conformation” is unclear. What is meant by catalytically grounded? And how can the authors rule out the much more like ensemble of conformational substrates, with the equilibria being shifted by individual mutations?

Response: We agree with the reviewer that we cannot rule out an ensemble of conformations during catalysis. To avoid confusion, we have removed the term ‘catalytically grounded’ on pages 10 and 12.

10. Line 342. What is the experimental basis of the conclusion that kinetically, “the Ct-extension exerts more influence towards product release rather than succinyl-CoA binding?” K_m cannot be equated with substrate affinity.

Response: We thank the reviewer for these comments and understand that our kinetic K_m values cannot be used to interpret substrate binding affinity. Our goal here was simply to state that previously published data of wild-type and XLP mutant enzymes using stopped-flow analysis (reference 9) fit our model. We have now reworded the text:

It is of note that others⁹ have shown that the C-terminal extension of hsALAS2 most significantly impacts ALA release after succinyl-CoA binding, suggesting why product release is the rate determining step.

Minor:

1. Line 29. Change “That has evolved among higher eukaryotes” to “that is only present in higher eukaryotes”.
2. Fig 1b. Monomer A described as being light green in the figure legend is shown in grey in the figure. Same problem in Fig. 1d, subdomain 2 and Fig 1e.
3. Lines 90/91 (Supplemental section). Change “that could accommodate for the disordered” to “that could account for the disordered”.
4. Line 184. Change “ π - π ” to “ π - π interactions”.
5. Line 191. FigS1 does not show the structure of succinyl-CoA bound ALAS2. Please correct.
6. Line 365. Define FECH.

Response: All minor points have now been corrected.

REVIEWERS' COMMENTS:

Reviewer #1 (Remarks to the Author):

The authors have addressed most of my comments satisfactorily. Two minor issues remain:

In my first point, I had indicated that in order to accommodate the substrate, the Ct-extension must already have moved in the simulation in which only PLP was bound. The authors now show that this is not the case, with the Ct-terminal in virtually the same position. What confused me is the sentence on page 9: "In order for succinyl-CoA to bind and orientate in the active site, we reason that the Ct-extension must change its conformation." But apparently, you were able to put it in and only then the Ct-terminal starts moving? Does this mean that there is overlap between the substrate and the Ct-terminal, leading to big forces and hence a distortion of the Ct-terminal? The authors say very little about how the substrate was modelled in. A distortion of the Ct-terminal can also just be the effect of not placing the molecules too carefully. If I understood correctly, the substrate was taken from the scALAS structure, in which the Ct-terminus is in an entirely different position. Could that also mean that the displacement of the Ct-terminal is only the onset of a much bigger movement, and it will eventually take a position similar to that seen in the scALAS structure? It would be good if they can explain also this aspect of the modelling effort in more detail.

The authors show the repeats of their simulations in their response to the reviewers, but in the supporting information, they only mention that figures S10 and S11 are a representative example of $n = 2$. What speaks against showing all data in the supplementary material? I think the data shows that there is an asymmetry, but also that there is a larger variability between the monomers and repeats.

Reviewer #3 (Remarks to the Author):

The authors were very responsive to this reviewer's comments. The Revisions are satisfactory and have strengthened the manuscript.

REVIEWERS' COMMENTS:

Reviewer #1 (Remarks to the Author):

The authors have addressed most of my comments satisfactorily. Two minor issues remain:

In my first point, I had indicated that in order to accommodate the substrate, the Ct-extension must already have moved in the simulation in which only PLP was bound. The authors now show that this is not the case, with the Ct-terminal in virtually the same position. What confused me is the sentence on page 9: "In order for succinyl-CoA to bind and orientate in the active site, we reason that the Ct-extension must change its conformation." But apparently, you were able to put it in and only then the Ct-terminal starts moving? Does this mean that there is overlap between the substrate and the Ct-terminal, leading to big forces and hence a distortion of the Ct-terminal? The authors say very little about how the substrate was modelled in. A distortion of the Ct-terminal can also just be the effect of not placing the molecules too carefully. If I understood correctly, the substrate was taken from the scALAS structure, in which the Ct-terminus is in an entirely different position. Could that also mean that the displacement of the Ct-terminal is only the onset of a much bigger movement, and it will eventually take a position similar to that seen in the scALAS structure? It would be good if they can explain also this aspect of the modelling effort in more detail.

Response:

The protein has enough space to accommodate the PLP in the active site. From the simulation with PLP, we did not observe any major Ct-extension movement. The same is not true with succinyl-CoA. In order to accommodate succinyl-CoA, the Ct-extension needs to move out and it was facilitated by salt bridge dynamics involving the residues in the Ct-extension. However, there was no distortion of the Ct-extension during modeling. We have explained the details below.

The substrates were modelled in the hALAS2 after aligning the *Rhodobacter capsulatus* ALAS (rcALAS) structure (which lacks a Ct-extension) with the hALAS2 structure. In the model generated thereafter, there were bad contacts between succinyl-CoA and the Ct-extension. However, to ensure that the Ct-extension does not undergo any distortion, the bad contacts were removed by performing minimisation and equilibration in the presence of restraints (a force with force constant $1000 \text{ kJ mol}^{-1} \text{ nm}^{-2}$) on the heavy atoms of the protein (for verification, the GROMACS input files for minimisation and equilibration were provided as Supplementary Software). This ensured that the Ct-extension retained its configuration during the removal of bad contacts. Thus, there were no bad contacts after the equilibration simulation, with the Ct-extension retaining its configuration.

The ability of the Ct-extension to accommodate succinyl-CoA stems from its inherent motions that arise as a result of the salt bridges E569-R511 and E571-(R293/K299) being transiently broken even in the absence of substrates. A look at Supplementary Fig. 9 can explain such behaviour of the protein in the absence of substrates. In addition to conformations where the salt bridges are intact (labelled "I" in the figure), the protein also exhibits conformations where these salt bridges are broken even in the absence of substrates (labelled "IIa" and "IIb" in the figure). It is this transient breaking of the salt bridges i.e. their dynamics that facilitates the binding of succinyl-CoA in the active site region.

The modelling of substrates in hALAS2 is now described in greater detail in the Methods section, pages 19-20.

The authors show the repeats of their simulations in their response to the reviewers, but in the supporting information, they only mention that figures S10 and S11 are a representative example of $n = 2$. What speaks against showing all data in the supplementary material? I think the data shows that there is an asymmetry, but also that there is a larger variability between the monomers and repeats.

Response:

We thank the reviewer for this suggestion, and have now included both sets of simulations in Supplementary Fig. 10 and 11.

Reviewer #3 (Remarks to the Author):

The authors were very responsive to this reviewer's comments. The Revisions are satisfactory and have strengthened the manuscript.